# A Novel Image Inpainting Method Used for Veneer Defects Based on Region Normalization

**DOI:** 10.3390/s22124594

**Published:** 2022-06-17

**Authors:** Yilin Ge, Jiahao Chen, Yunyi Lou, Mingdi Cui, Hongju Zhou, Hongwei Zhou, Liping Sun

**Affiliations:** College of Mechanical and Electrical Engineering, Northeast Forestry University, Harbin 150040, China; gylgyl777@163.com (Y.G.); 15604615447@163.com (J.C.); yylhtoh@163.com (Y.L.); mingdi1998@nefu.edu.cn (M.C.); zhouhongjv@163.com (H.Z.); easyid@163.com (H.Z.)

**Keywords:** image inpainting, veneer defect, region normalization, hybrid dilated convolution

## Abstract

The quality of the veneer directly affects the quality and grade of a blockboard made of veneer. To improve the quality and utilization of a defective veneer, a novel deep generative model-based method is proposed, which can generate higher-quality inpainting results. A two-phase network is proposed to stabilize the network training process. Then, region normalization is introduced to solve the inconsistency problem between the mean and standard deviation, improve the convergence speed of the model, and prevent the model gradient from exploding. Finally, a hybrid dilated convolution module is proposed to reconstruct the missing areas of the panels, which alleviates the gridding problem by changing the dilation rate. Experiments on our dataset prove the effectiveness of the improved approach in image inpainting tasks. The results show that the PSNR of the improved method reaches 33.11 and the SSIM reaches 0.93, which are superior to other methods.

## 1. Introduction

Solid wood panels are natural and eco-friendly materials, which are popular with consumers as decorations. Nevertheless, with the decrease in forest resources, developing the wood-based panel industry will promote forest-based industry without damaging forestry. As a kind of veneer wood-based panel, blockboard consists of veneers bonded to center layers and is widely used in furniture manufacturing and interior design because of its minimal glue consumption and low cost [1]. In order to increase the environmental protection of blockboard and reduce the waste of materials, Nazerian et al. [2] and Teixeira et al. [3] improved the center layers of the blockboard, but ignored the quality of the veneers on both sides of the center layers. In recent years, consumers have paid more attention to product appearance and the consistency of the surface texture, but during the growth and processing of wood different types of defects may appear, which will negatively affect the performance and ornamental value of the wood [4,5]. Therefore, in order to maintain the consistency of the surrounding texture, optimizing the quality and the utilization of a defective veneer requires urgency. A novel method is applied in image inpainting technology to identify and inpaint defects in the surroundings areas of a veneer. This enables the inpainted areas to mimic the nature of the veneer. This approach then identifies the taped patches with similar color, size and texture to restore the affected areas close to their original form. This improves the quality and the utilization of veneers, which eventually reduces the yield of waste veneers and prevents the overuse of natural resources.

Image inpainting is a classical topic of research in computer vision. The purpose of image inpainting tasks is to restore the missing regions by means of the content of the known areas, thus creating the missing parts with plausible accuracy. Deep learning methods have achieved good results in the field of image inpainting. Some recent studies effectively used the information around the missing regions to generate better results. These methods can be divided into two patterns. The first pattern proposes partial convolution [6,7] to handle the color inconsistency near the hole boundaries by only operating on valid pixels. However, these methods fail to address the semantic fuzziness and often cause a shift in the mean and variance. The second pattern demonstrates that dilated convolution [8,9] can enlarge the receptive field without decreasing the resolution of the image. Dilated convolution may obtain more scale context information and achieve a better prediction of missing areas. Iizuka et al. [10] applied dilated convolution to reconstruct face images, and Van et al. [11] employed dilated convolution to inpaint natural images. However, as mentioned in [12], when the rate of dilated convolution increases, much local information is lost, thus resulting in the gridding effect.

To achieve a better image inpainting effect, region normalization (RN) is proposed to normalize the missing regions and known regions separately to solve the mean and variance shift problem. Meanwhile, a simple hybrid dilated convolution (HDC) module is applied, which uses a different dilation rate for each layer to expand the receptive field of the network, thereby reducing the gridding effect. The improved model are divided into two parts to make the network more stable. The first part can rough out the missing regions by training a rough network. A detailed network with the HDC module further refines the content of the rough predictions. In addition, RN is applied for both the rough network and the detailed network. The improved network can achieve high-quality inpainting results compared with other methods. The proposed method can complete masked panels in different locations and sizes and generate a more coherent surface texture.

Our specific improvements are as follows:Taking into account the impact of spatial distribution on normalization, region normalization is introduced to divide pixels into different regions before calculating the mean and standard deviation of each area. Region normalization significantly enhances network performance.The HDC module is introduced to the detailed network to reconstruct the defective areas of the wood by changing the expansion rate, leading to a continuous texture with exquisite detail.The modified novel method in terms of validity and generativeness and demonstrate that the improved network can obtain satisfactory performance results in image inpainting and can not only restore the texture of veneers but also generate the defective regions of veneers.

## 2. Related Works

### 2.1. Image Inpainting

In recent years, much research has been conducted on image inpainting. The available image inpainting approaches can be divided into traditional and deep learning approaches. Traditional approaches [13,14,15,16,17,18,19] aattempt to find patches from the background region to restore the hole. These methods only produce better results on images with simple cases, but the effectiveness becomes worse when handling images with complex texture and large missing areas. However, veneer is a natural material, and its surface characteristics are very complex, with random texture properties. Therefore, if an image of a veneer defect with complex texture is reconstructed by traditional methods, the generated contexts always show discontinuous pixels.

With the development of deep learning in computer vision technology, great progress has been made in the field of image inpainting. Using deep learning to reconstruct veneer defects not only overcomes the shortcomings of traditional methods, but also ensures the consistency of the texture and color of the generated image. In 2016, Pathak et al. [20] proposed context-encoder network structures and the GAN [21] network to restore missing street scene images. The structure of the context encoder learns the features of the global image and then infers the missing part. The GAN network is applied to judge the difference between the predicted image and the real image. However, this structure can only handle fixed-size images, and is difficult to train against loss when the input image becomes large. Yang et al. [22] introduced two networks to overcome this problem in which the content generation inferred the possible content of the missing part, and the texture generation was used to enhance the texture generated by the content network. The results of the context encoder are then input to gradually increase the texture detail. However, the optimization process of this approach significantly increases the calculation cost. Yu et al. [23] proposed partial convolution and then concurrently employed the SN-Patch GAN discriminator to achieve better predictions. However, this approach does not explicitly consider the correlation between effective features, which may produce semantic errors in the generated content. To make the generated images more completed and the texture more finely detailed, Yu et al. [24] proposed a novel contextual attention layer which learned characteristic contents from relevant background patches to produce missing patches. Yan et al. [25] introduced the shift-connection layer to the U-Net [26] layer to infer the relationship between the relevant holes in the context area. Although the above methods can produce realistic results, they often lead to inconsistent colors and discontinuous pixels. Some studies have shown that normalization plays a key role in image-restoration tasks, but none of the existing methods have normalized the missing regions separately.

### 2.2. Normalization

In order to improve the model’s convergence speed, normalization is added to the deep learning technology for image inpainting tasks. There are currently several normalization approaches: batch normalization (BN) [27] is a normalization approach with a superior effect when compared with similar methods. However, the effect of batch normalization is not ideal when the batch size is small, as the mean and variance obtained in the calculation process cannot represent the overall situation. Instance normalization (IN) [28], meanwhile, is not affected by the channel or batch size, but if there is a correlation between the feature map and the channel, then this form of normalization is not recommended. Switchable normalization [29] combines the above normalization approaches, but the training is very complicated and time-consuming. Unlike most visual tasks, image inpainting tasks are divided into two regions: known regions and missing regions. The existing methods neglected the missing areas, leading to shifts in the mean and variance of normalization. In order to avoid these shifts, the modified method normalize the mean and variance of each area to improve the quality of the inpainting network.

## 3. Approach

Iizuka et al. [10] showed advanced results when inpainting images of faces, buildings and other tasks. However, veneers are different from faces and buildings, with the texture of veneers being of different directions and types. Therefore, by improving the inpainting model [10] the improved network can achieve better results in the field of veneer inpainting. The overall framework of the improved method can be seen in Figure 1. The input is an irregular veneer image with masked area, which then outputs the reconstructed image. In image inpainting tasks, a larger receptive field can capture more scale context information, and improve the inpainted image’s visual consistency. To increase the receptive field and create a more stable training process, the improved model consists of two networks. Let $I_{in}$ be the input of the rough generator: during the rough inpainting process, we achieve the rough prediction $I_{p}$.Then, the detailed network with an HDC block takes the $I_{p}$ and $I_{in}$ to output the final result $I_{f}$. The final result If approaches the original image $I_{gt}$. Global and local context discriminators are used to distinguish the ground-truth image from the completed image. Focusing the three networks together can generate a final completed image of veneers. Although the task is divided into two parts, the whole network can be trained in an end-to-end manner.

### 3.1. Rough Inpainting

The rough network is given in Figure 2. The input of the rough network is an irregular veneer image with missing region, which connects the features of each layer of the encoder and decoder layer; converts the feature map into RGB with the same resolution as the input; and gradually restores the accuracy of the image. In the encoder, 3×3 convolutions are used with a step size of 2. Leaky Relu and Basic Region normalization are applied in each convolutional layer, and Relu and Later Region normalization are used in each convolutional layer in the decoder. The reconstruction of losses is then used to train a rough network.

### 3.2. Detailed Inpainting

#### 3.2.1. Detailed Network

The architecture of the detailed network is shown in Figure 3 To achieve the final result $I_{f}$, the predicted pictures $I_{p}$ and $I_{in}$ are inputted through a rough inpainting process to the detailed network. In the encoder, each layer is composed of a 3×3 convolution and a 4×4 dilated convolution, and an attention module is then added to the third layer of downsampling, which is fused with its feature map. The bottleneck block is composed of four stacked HDC modules. In the encoder and the HDC module, leaky Relu and basic region normalization are applied for each convolutional layer, and Relu and later region normalization are used for each convolutional layer in the decoder.

#### 3.2.2. Hybrid Dilated Convolution

Iizuka et al. [10] used dilated convolution to increase the convolution’s receptive field without decreasing the resolution of the image. As shown in Figure 4a, all convolutional layers have a dilation rate, r, of 2. A pixel in the second layer uses nine pixels from the first layer, then a pixel in the third layer uses nine pixels from the second layer, which is equivalent to 25 pixels from the first layer. However, in the fourth layer, many of the pixel values in dilated convolution are not utilized, causing discontinuous convolution kernels and the gridding effect. Unlike the dilated convolution, as shown in Figure 4b, hybrid dilated convolution (HDC) changes the dilation rates to be 1, 2 and 3, by setting different rates, which ensures that any pixel on a high level uses low-level data in continuous areas. When the rate is equal to 1, it preserves the complete 3×3 region to avoid losing the underlying information, and the rate settings of the subsequent layers are just enough to ensure the coherence of the receptive field. When HDC and dilated convolution are identical in terms of the number of parameters, convolution kernel size and the same receptive field, HDC can obtain more available information, thus reducing the gridding problem. The structure of the HDC block is shown in Figure 5.

The receptive field [30] is defined as:(1)$$l_{k}=l_{k-1}+\left[\left(f_{k}-1\right)*\prod_{i=1}^{k-1}s_{i}\right],$$
where $l_{k-1}$ corresponds to the receptive field of layer $l_{k-1}$, and $f_{k}$ is the convolution kernel size of the *k* layer.

#### 3.2.3. Attention Block

The attention mechanism can focus on the local information of the image, select the interested part and suppress useless information, then infer the content of unknown regions from the known regions. As shown in Figure 6, in order to generate the unknown region *U*, the attention module selects 3×3 patches $k_{i}\left(i\in \left[0,n\right]\right)$ from the known region *K*, then reshapes them as convolution kernels to compute similarity with the unknown region by cosine similarity.
(2)$$S\left(u_{i},k_{i}\right)=\frac{u_{i}k_{i}}{∥u_{i}{∥}_{2}{∥k_{i}∥}_{2}}$$
*S* represents the similarity of the known region $k_{i}$ and the unknown region $u_{i}$.

Then, softmax is applied to weigh the similarity in order to obtain the attention score for each pixel.
(3)$$A\left(u_{i},k_{i}\right)={\mathrm{softmax}}_{u_{i}}\left(wS\left(u_{i},k_{i}\right)\right)$$
where *w* is a constant value.

### 3.3. Region Normalization

Since the image inpainting task is divided into missing regions and known regions, here borrowing the idea in [31], the BN applied in [10] is improved, and the image inpainting task is normalized by region normalization. The pixels are divided into different areas, and the mean and standard deviation of each area are computed to achieve normalization. We suppose that the input feature $X\in A^{N\times C\times H\times W}$, and that *N*, *C*, *H* and *W* are the number, channels, height, and width, respectively. The equation is as follows:(4)$$X_{n,c}=R_{n,c}^{1}\cup R_{n,c}^{2}\cup \cdots \cup R_{n,c}^{m},$$
where $X_{n,c}$ means dividing the *c*-th channel of the *n*-th feature map into multiple areas.
(5)$$\mu_{n,c}^{m}=\frac{1}{\left|R_{n,c}^{m}\right|}\sum_{x_{n,c,h,w}\in R_{n,c}^{m}}x_{n,c,h,w},$$
here, *m* is a region index, and $\left|R_{n,c}^{m}\right|$ is the number of pixels in region $R_{n,c}^{m}$. The normalization of each region is calculated as:(6)$$\sigma_{n,c}^{k}=\sqrt{\frac{1}{\left|R_{n,c}^{m}\right|}\sum_{x_{n,c,h,w}\in R_{n,c}^{m}}{\left(x_{n,c,h,w}-\mu_{n,c}^{m}\right)}^{2}+\varepsilon }.$$

The mean and standard deviation of each small area are shown below:(7)$${\hat{R}}_{n,c}^{m}=\frac{1}{\sigma_{n,c}^{m}}\left(R_{n,c}^{m}-\mu_{n,c}^{m}\right),$$
(8)$${\hat{X}}_{n,c}={\hat{R}}_{n,c}^{1}\cup {\hat{R}}_{n,c}^{2}\cup \cdots \cup {\hat{R}}_{n,c}^{m}.$$

Finally, the normalization of each small area is combined together.

Region normalization is divided into basic region normalization (RN-B) and later region normalization (RN-L). RN-B is shown in Figure 7a. RN-B is generally used in the early layers to normalize the unmasked and masked areas separately to address serious mean and variance shifts. However, after multiple convolutions, the unmasked and masked areas are fused together, and if a region mask is futher used, this method becomes less effective. Then, to address this problem, RN-L is applied to produce a region mask by using the spatial response of the input features to detect masked regions. RN-L is shown in Figure 7b. Maxpool and avgpool are used to obtain two feature maps, which are connected together, and then the sigmoid function is used to obtain a spatial response map. The last region mask is obtained when the threshold t=0.8. RN-L is often applied in the later layers.

### 3.4. Loss Functions

In [10], the authors rely on GAN using Jensen–Shannon (JS) to weigh the distance between the produced distribution and the real distribution, but when the two distributions do not overlap, the JS divergence is zero and the gradient disappears. The improved Wasserstein GAN–Gradient Penalty (WGAN-GP) relies on the Wasserstein distance to weigh the distance between two distributions. The loss can show the distance even though the two distributions do not overlap.

WGAN-GP also proposes a new Lipschitz continuous restriction technique with gradient penalty, which solves the problem of training gradient explosion.

The Wasserstein distance [32] is defined as:(9)$$W\left(P_{r},P_{g}\right)=\underset{\gamma ∼\prod \left(P_{r},P_{g}\right)}{\inf}E_{(x,y)∼\gamma }[∥x-y∥].$$

The gradient penalty [33] is defined as:(10)$$\lambda E_{\hat{x}∼P_{\hat{x}}}\left[{\left({∥\nabla_{\hat{x}}D\left(\hat{x}\right)∥}_{2}-1\right)}^{2}\right].$$

To make the image semantic and to generate a more coherent texture, the following loss functions are used to optimize the network during the training process: pixel reconstruction loss and WGAN-GP [32,33] loss. In the generator, the pixel reconstruction loss is applied to both rough and detailed networks. The pixel reconstruction loss weighs the pixel-wise difference between the generated content and its corresponding original content, which produces less image blurring. Two training steps are performed with $L_{1}$ distance as a pixel reconstruction loss to make the results $L_{re1}$ and $L_{re2}$ as close as possible to the ground-truth image.
(11)$$L_{re}={∥I_{p}-I_{in}∥}_{1}+{∥I_{f}-I_{in}∥}_{1},$$
(12)$$L_{re1}={∥I_{p}-I_{in}∥}_{1},$$
(13)$$L_{re2}={∥I_{f}-I_{in}∥}_{1}.$$

$L_{1}$ loss is widely used in image inpainting tasks, as it has high stability and makes the texture more detailed. The data created by the generator is therefore closer to the true distribution, and pixel reconstruction loss can reconstruct high-quality wood textures.

In the discriminator, the improved WGAN-GP is used to replace other GAN transformation losses.
(14)$$L_{d}=-E_{x∼p_{\gamma }}\left[D\left(x\right)\right]+E_{x∼p_{g}}\left[D\left(x\right)\right]+\lambda E_{\hat{x}∼p_{\hat{x}}}{\left[{∥\nabla_{x}D\left(x\right)∥}_{p}-1\right]}^{2},$$
(15)$$L_{D}=L_{d}\_\mathrm{global}+L_{d}\_\mathrm{local},$$
(16)$$L=\lambda_{g}L_{re}+\lambda_{d}L_{D},$$
where D(x) denotes the discriminator network containing the parameter, and λ is the gradient penalty weight factor.

## 4. Results and Discussion

The original images are obtained by using the image acquisition equipment Oscar F810 CIRF, which is produced by the Allied Vision Technologies company in Germany. We shoot 7500 images of wood veneer, using a high-density white LED light array at a lower angle, so that the collected images can truly reflect the information of the veneer. Then, the captured photos are cropped to 200×200 pixels and divide the images into defective and non-defective images. To verify the performance of the model in processing the defects on the surface of wood veneer, traditional data enhancement techniques are applied during the training process: the images are rotated, cropped and flipped in terms of spatial transformation, and in the aspect of color distortion, the images are enhanced by changing the brightness and tone. Finally, 40,000 images are used for training, while 4000 panel images are used for testing. The experimental model is based on Pytorch [34], which is trained on one GPU: NVIDIA 2080TI.

### 4.1. Analysis of Effectiveness Results

In image inpainting tasks, the peak signal-to-noise ratio (PSNR) and structural similarity index (SSIM) [35] evaluation metrics are usually used to measure the performance of the model. A comparison is made between the improved method and the current state-of-the-art methods, GL [10] and GntIpt [24], using our dataset. The improved ap-proach performed the comparison on the central mask, and Table 1 demonstrates that the improved approach outperformed other methods in this area. In addition, Table 2 show the effectiveness of the improved method in different sizes.

### 4.2. Analysis of Generative Results

Figure 8 and Figure 9 show the inpainted results on both center and irregular holes. To compare the improved method with GL [10] and GntIpt [24] using our dataset. For the center mask, as shown in Figure 8, the effect of GL [10] is unsatisfactory, and the reconstructed results are to some extent blurry. GntIpt [24] shows a better effect due to the globally and locally consistent adversarial network, but its predictions still have a gridding effect and lack of detail to a certain extent. For irregular masks, as shown in Figure 9, when the missing area is small, all methods can generate a plausible and smooth inpainting result. However, when the missing area is large, the predictions of GL [10] are blurred and the textures are discontinuous; GntIpt [24] can produce smooth textures, but its predictions lack detail. The improved method can solve these problems, effectively dealing with the details. The improved network can eliminate the influence of mean and variance deviation on training, and plays a decisive role in eliminating the gridding effect.

### 4.3. Ablation Studies

#### 4.3.1. The Effect of RN

We investigate the effectiveness of RN in the inpainting model and compare the results with those of the instance normalization (IN) method. As shown in Figure 10, the result performances are distorted when using IN, and RN is more effective in image painting.

#### 4.3.2. The Effect of HDC Layer

As is obvious in Figure 11, directly using a conventional 3×3 layer leads to the failure of inpainting to restore the limpid texture, and the gridding phenomenon of the reconstructed image is obvious when using the conventional method. In contrast, the improved method can improve the performance, and the gridding effect is significantly reduced. This illustrates that semantic coherence is constructed by the HDC layer.

### 4.4. Reconstruction Experiments on Defective Regions

The above experiments prove that the improved method is effective in restoring the texture of the veneers. We conduct further experiments to reconstruct defective regions of different positions, sizes, and quantities.

#### 4.4.1. Reconstruction on Veneer Defective Areas

Figure 12a shows no texture around the defective area, whereas Figure 12b shows textures around the defective area. This proves that the improved method can achieve good results in the reconstruction of defective areas, and could potentially meet the requirements for eliminating defects in a veneer.

#### 4.4.2. Generation of Different Numbers of Defective Regions

The defective regions on the surface of the veneers are not unique. To verify the multi-effectiveness of this network, the images with single and two defective regions are inpainted on the veneers separately. As shown in Figure 13, the results of the improved method are semantically consistent and visually consistent for inpainting veneers with both single and two defective regions.

## 5. Conclusions

We propose a novel rough-to-detailed deep generative framework with region normalization and an HDC module, which significantly improves image inpainting results. The proposed network can successfully inpaint the defective veneers and reconstruct the texture of the veneers, thereby improving the grade of the blockboard. The improved model can process holes of any shape, size, and location. However, one limitation of the method is that it fails to deal with the largest holes. The focus of the future work is to improve the network to inpaint the largest holes and identify the veneer patches to match the predicted results. 

## Figures and Tables

**Figure 1 sensors-22-04594-f001:**
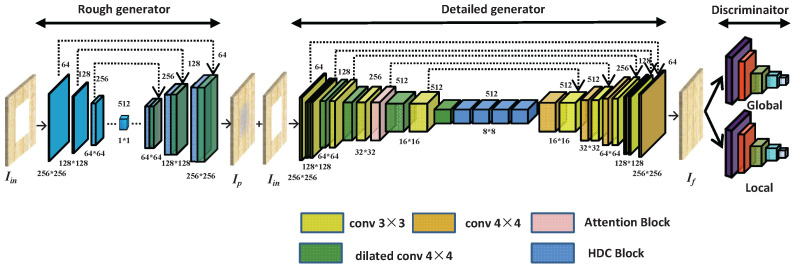
The architecture of the improved model.

**Figure 2 sensors-22-04594-f002:**
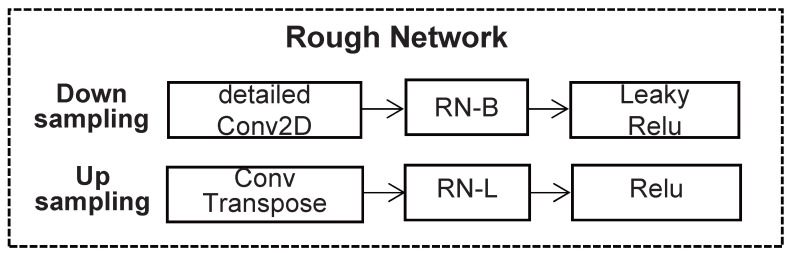
Architecture of the rough network.

**Figure 3 sensors-22-04594-f003:**
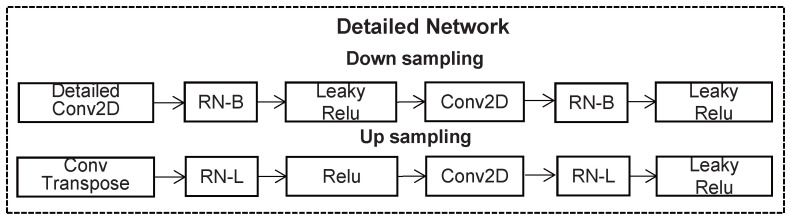
Architecture of the detailed network.

**Figure 4 sensors-22-04594-f004:**
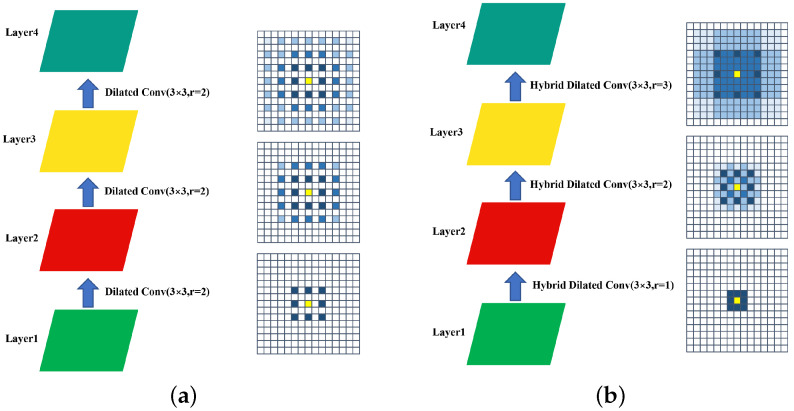
Illustration of the gridding problem. Blue marked pixels represent the number of times each pixel of the first layer has been used by the current layer; the larger the number, the deeper the color. (**a**) Dilated convolution: All convolutional layers have a same dilation rate, *r*, of 2. (**b**) Hybrid dilated convolution: The dilation rates of subsequent convolutional layers are 1, 2, and 3, respectively.

**Figure 5 sensors-22-04594-f005:**
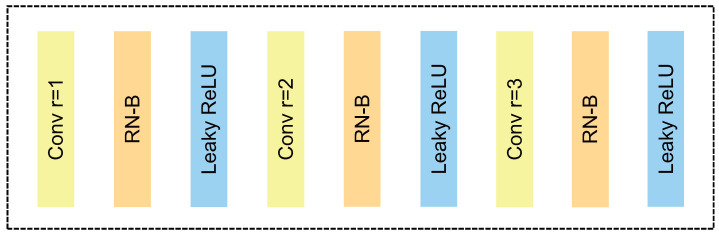
The structure of the HDC block. The HDC block consists of three convolution layers with a kernel size of 3 and dilation rates of 1, 2, and 3, respectively.

**Figure 6 sensors-22-04594-f006:**
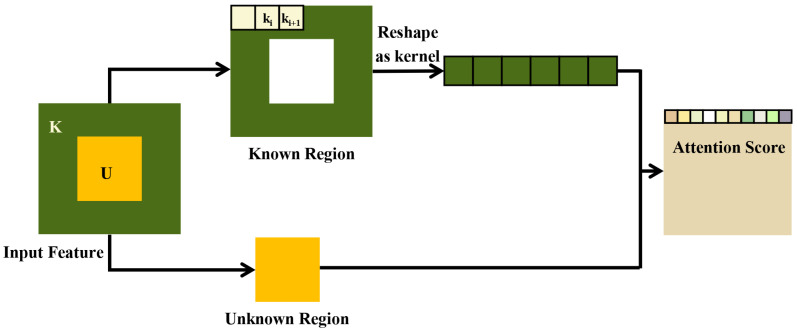
Illustration of the attention layer.

**Figure 7 sensors-22-04594-f007:**
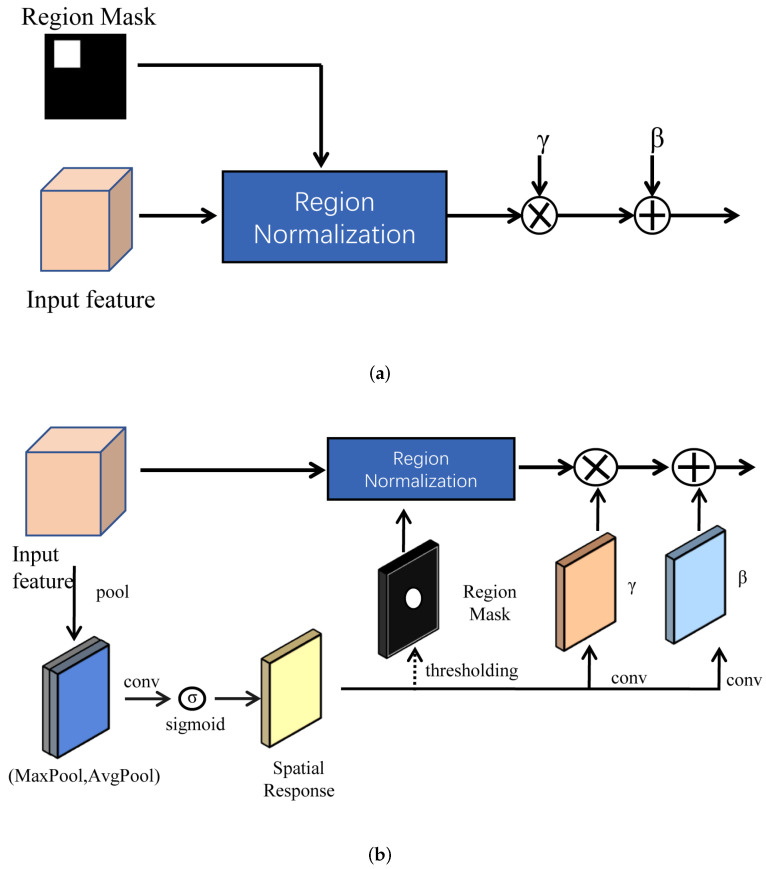
Architecture of RN (**a**) Architecture of RN-B (**b**) Architecture of RN-L.

**Figure 8 sensors-22-04594-f008:**
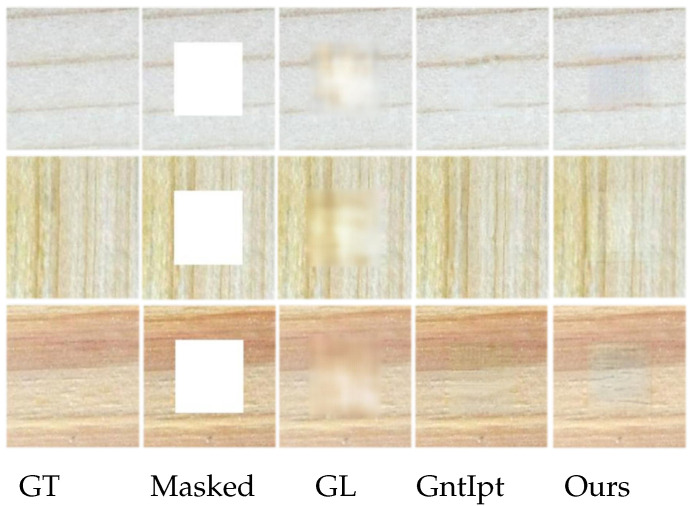
Qualitative comparisons in center mask cases.

**Figure 9 sensors-22-04594-f009:**
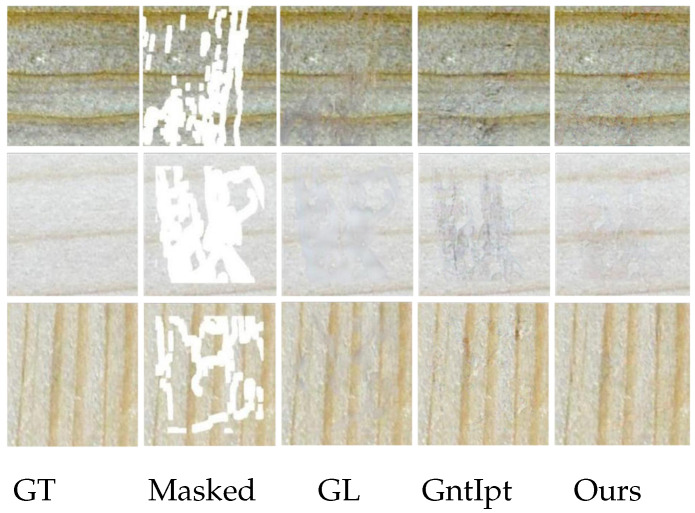
Qualitative comparisons in irregular mask cases.

**Figure 10 sensors-22-04594-f010:**
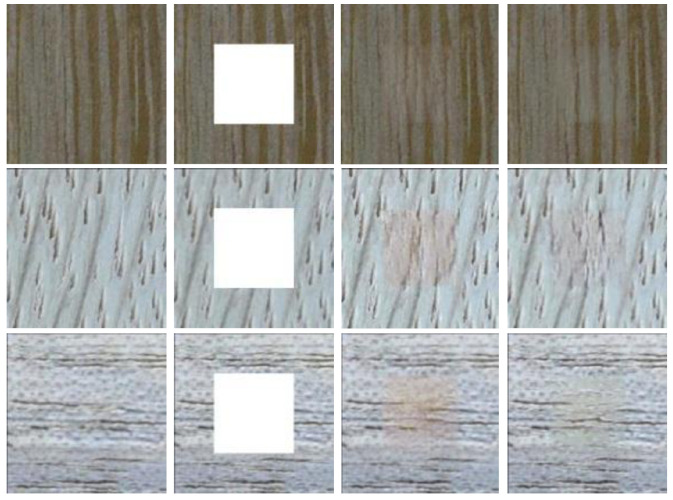
The effect of region normalization. From left to right are shown the original image, the input image, an image with IN, and our result, respectively.

**Figure 11 sensors-22-04594-f011:**
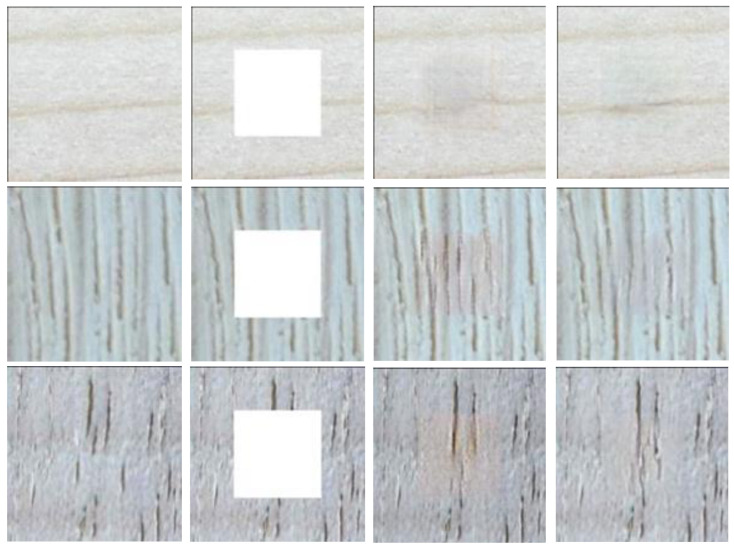
The effect of the HDC module. From left to right are shown the original image, the input image, an image without an HDC moudle, and our result, respectively.

**Figure 12 sensors-22-04594-f012:**
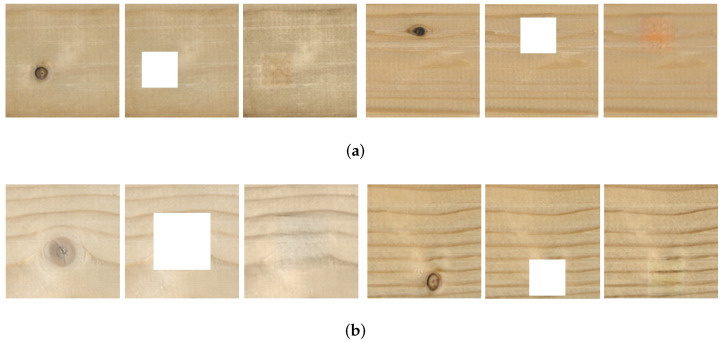
Reconstruction of defective areas on a veneer (**a**) No texture around the defective area (**b**) Textures around the defective area.

**Figure 13 sensors-22-04594-f013:**
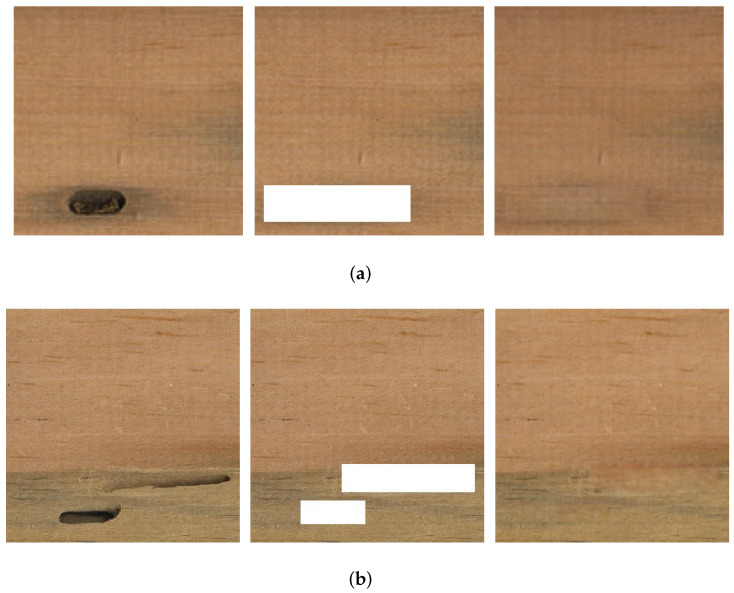
The generation of different numbers of defective regions (**a**) The result of single defective region on veneers. (**b**) The result of double defective regions on veneers.

**Table 1 sensors-22-04594-t001:** Effectiveness results and comparisons of center holes.

Dataset	GL	GntIpt	Ours
PSNR	27.45	30.22	33.11
SSIM	0.86	0.90	0.93
MSE	0.11	0.059	0.049

**Table 2 sensors-22-04594-t002:** Effectiveness results of irregular holes.

	PSNR	SSIM	MSE
10–20%	36.61	0.969	0.000218
20–30%	33.26	0.937	0.000473
30–40%	32.11	0.912	0.000615
40–50%	29.05	0.901	0.001244

## Data Availability

The datasets of the current study are available from the corresponding author upon reasonable request.

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
