# Peer review of "A Novel Image Inpainting Method Used for Veneer Defects Based on Region Normalization"

_sensors, 2022, doi:10.3390/s22124594_

Round 1

Reviewer 1 Report

Dear Authors

Although the manuscript is quite interesting in terms of results, concept and output, it requires revision to fit the journal requirements for publication. The language of the paper is not OK.

Scientifically, the paper is good, it has main drawback in writing and presenting the concept.

I do recommend the authors to address the reviewer’s concerns in the revised version. Regarding the figures, the number of figures must be reduced.

  1. The language of the paper must be reviewed and revised. It is not written in a proper and comprehensive order. Please consider this major revision in the revised version. For instance, it is recommended to use the passive tense in the academic manuscript. Therefore, please avoid and remove the active tense starting with “WE”, “OUR”. Please consider this in the revised version.
  2. Please reduce the number of subsections; why did you define this kind of structure? It is not a report; it is a paper. It must be comprehensive not excessive!
  3. Please stress the novelty of the work in the abstract. Due to the title, it is expected to clearly explain the novelty of the work.
  4. Please add a table of nomenclature.
  5. Regarding the mathematical developments, please cite a reference for them.
  6. About the threshold stating after figure 6, how did you arrive at this value? Have you thought about other values? Please comment on this matter.
  7. Section 3.3, it was stated “…3 ∗ 3 region was considered to avoid losing the underlying information” how did you arrive at this concept? Why 3 by 3?
  8. Too many figures! Why? Please merge or simplify them!
  9. Conclusions are not supportive! Please rewrite them in a way that they present the main outcome of the work, application, advantages/disadvantages and any other important aspects.

Very Best

The Reviewer

Author Response

Dear Reviewer: 

Thank you very much for your insightful comments concerning our manuscript entitled “A novel of image inpainting method for veneer defect based on Region Normalization”. Based on your comment and request, we have made extensive modification on the original manuscript. A document answering every question was enclosed. Please see the attachment.

Best Regards,

Authors.

Reviewer 2 Report

The topic is great.

I liked the Region Normalization method because enhances network performance, however the model can obtain performance results in image inpainting when compared to other methods. References are ok.

There are some little problems with the English in some sentences. Authors must to check this point.

Author Response

Dear Reviewer: 

Thank you very much for your insightful comments concerning our manuscript entitled “A novel of image inpainting method for veneer defect based on Region Normalization”. Based on your comment and request, we have made extensive modification on the original manuscript. Please see the attachment.

Best Regards,

Authors.

Round 2

Reviewer 1 Report

Dear Authors

Your efforts to revise the paper is highly appreciated, now, The Reviewer is agreeing with its publication at this stage.

Very best regards

Good Luck

The Reviewer